# Irrigation and inequalities in the Senegal River Delta

**El Hadji Malick Sylla**[1]*, **Boubacar Ba**[2], **Khalifa Diop**[3], **Bruno Barbier**[4],
**Sidy Mohamed Seck**[2], **Patrick Van Damme**[5]

1 African Population and Health Research Center, West Africa Regional Office (WARO), Dakar, Senegal,
2 Université Gaston Berger de Saint-Louis, Saint-Louis, Senegal, 3 Université Numérique Cheikh
Hamidou Kane, Dakar, Senegal, 4 Centre de Coopération Internationale en Recherche Agronomique pour
le Développement (Cirad), Montpellier, France, 5 Université de Gent (Belgique) – Czech University of Life
Sciences, Faculty of Tropical AgriSciences (Czech Republic), Czech Republic

* msylla@aphrc.org

## Abstract

### Introduction

Irrigation is considered to be one of the best solutions to fight drought spells in agriculture but also to lift farmers out of poverty through improved water provision, and thus crop yields. However, irrigated agriculture, as organized in Africa, has often generated new inequalities between farmers because of the larger areas cultivated by a minority of farmers, which allows them to earn higher incomes compared to small farmers. This study aims to provide evidence from the Senegal river delta, one of the largest irrigated rice areas in Senegal, on how inequalities have indeed increased over time.

### Methods

It is based on a survey conducted among 326 households (HHs) on total land area cultivated, investment in irrigation and agricultural income. Inequalities were calculated through Lorenz curve and Gini index.

### Results

Results show the large disparities in areas cultivated by each HH. A minority cultivates more than 100 ha while the majority cultivates less than 2 ha. As a result, only some 7% of producers generate profits from irrigated agriculture, while most HHs experience net income deficits. In addition, due to difficulties in applying double-cropping of rice, especially among small farmers, inequalities remain what they are, while the hydro-agricultural developments carried out in this area were supposed to allow even three crops per year on the same plot.

journal.pone.0325862

University: Universidad Veracruzana, MEXICO

**Peer Review History:** PLOS recognizes the
benefits of transparency in the peer review
process; therefore, we enable the publication
of all of the content of peer review and
author responses alongside final, published
articles. The editorial history of this article is
available here: https://doi.org/10.1371/journal.
pone.0325862

**Data availability statement:** All materials and data used to produce the results presented in the paper are included as Supporting information files.

**Funding:** The author(s) received no specific funding for this work.

**Competing interests:** The authors have declared that no competing interests exist.

## Conclusion

As there is not enough idle land left, it is difficult to increase the areas planted by HHs. In the absence of new developments, the practice of double-cropping on existing farms, especially among small farmers, would help reduce inequalities and facilitate rice self-sufficiency, which the country has been seeking for decades.

## Introduction

Agriculture is globally the main sector that can help to achieve United Nations Sustainable Development Goals 1 and 2 [1]. It can also be an important lever for reducing poverty [2] and inequality in developing countries [3]. However, it contributes less and less to the latter's Gross Domestic Product, but remains the major job-creating sector and of course the main contributor to food security in most of them [4]. Although agricultural production is growing in almost every developing country, its yield increases contribute insufficiently to fight poverty. Around 80% of the world's poor live in rural areas and 64% of them work in agriculture [5]. Increasing agricultural productivity is a necessary condition for eradicating poverty in rural areas of the world [6].

Irrigation is often presented as the central solution to a lot of agricultural problems. If well managed, it increases productivity, reduces crop failure risks, creates direct and indirect agricultural employment, allows for crop diversification, increases cultivated areas especially in lowlands, reduces poverty and helps reduce deforestation [7]. It benefits a lot from funding from states and international organizations who consider it as one of the essential levers to the many current and future challenges of agriculture.

Paradoxically, irrigation has mixed effects. Irrigated crops are vulnerable to heat waves and post-harvest losses, both of which contribute to the high operating and maintenance costs of irrigation equipment [8,9]. Irrigation can also become a factor promoting or exacerbating inequality between farmers because of the unequal distribution of land and water [10]. Case studies from rural India and other LMICs show that poverty is mainly due to unequal distribution of land, water and value added, which are themselves explained by unfair social dependency relations rooted in tradition [11].

In the Senegal River Delta, the state has been trying since its accession to independence in 1960 to develop irrigated rice to achieve rice self-sufficiency, reduce poverty and balance the trade deficit. Therefore, several irrigated perimeters were built. In the 1970s and 1980s, the Senegal river valley moved from traditional recession agriculture to irrigated agriculture with complete water control involving Asian rice (*O. sativa*), tomato and onions. This was followed by the establishment of Private Irrigated Areas in the 1990s, and, from 2004 onwards, the promotion of Public-Private Partnership projects [12]. In the given climatic conditions, irrigation potentially allowed for three crops per year. Rice can be grown in the hot dry season (March-June), or during the cold season (November – February) and during the rainy season

(July-october), whereas vegetables (onions, tomatoes, but also pepper or okra) can be grown during the rainy season and in the cold off-season.

Studies of farms in the Senegal River Delta have shown significant disparities in access to land and agricultural inputs. Poor farming households (HHs) often lack land to start with and generate insufficient income [13]. Although the expansion of export- or local processing oriented cropping, mainly focussing on tomatoes, has reduced poverty in the Delta more quickly than elsewhere in Senegal [14], its development, which mainly benefits large producers, penalizes small farmers, particularly young people, who struggle to access (irrigated land) and turn to precarious employment or even migrate [15]. Historical public policies implemented after independence, the structural adjustment plan of the 1990s and the public-private partnership of the early 2000s have exacerbated pre-existing inequalities in the Delta region [16].

The hypothesis underlying the present analysis is that the inequalities observed among irrigated farmers in the Senegal River Delta can be explained, in part, by the small areas cultivated and the limitation of the majority of farmers to a single crop campaign, whereas ideally irrigated agriculture in the area allows to have three crop campaigns per year. The validity of this hypothesis could guide new policy directions in Senegalese agriculture. Even though Senegal has, since 1960 only been self-sufficient once, the national government wants to achieve food sovereignty by 2029 [17]. It will, however, be necessary to take up the lessons learned from the evolution of irrigation in the Delta in order to understand how irrigated areas are best developed and how income is generated and can be equally distributed over producers. Research results will make it possible to better guide State and donors interventions in reducing inequalities in the area. However, only a few studies have addressed these aspects.

## Materials and methods

The Senegal river Delta is characterized by a semi-arid Sahelian climate. Agricultural activities are strictly dependent on irrigation from the Senegal River. The predominant irrigation method is flooded irrigation via a network of canals managed by SAED. The soils are mainly vertisols and gley soils, known locally as *hollaldé*, which have a high clay content (>40%), ensuring excellent water retention for rice but requiring high mechanical energy for soil preparation. Average water consumption for a rice cycle ranges from 10,000–12,000 m³/ha. These high technical requirements, combined with high energy costs for pumping, constitute a significant entry barrier for smallholders. The study was conducted in the Central and Lower Delta areas, corresponding to the municipalities of Diama and Ross Béthio (Fig 1). The latter have benefited from all types of hydro-agricultural developments carried out in the Delta.

In these two municipalities, localities surveyed were selected based on geographical position in relation to the land and irrigated areas, the ethnic group and relative demographic weight. In total, 326 HHs spread across 20 villages were surveyed. The 20 villages were selected using a stratified purposeful sampling approach. From the comprehensive list of settlements in the Diama and Ross Béthio districts provided by the SAED local delegations and the municipalities, we stratified villages based on their demographic size and ethnic composition (to ensure representation of Wolof, Peulh, and Moor communities). Within each stratum, villages were selected to cover the geographic diversity of the Delta.

In each village, a simple random selection was carried out within a stratified sample of producers, taking into consideration the number of households as well as the distinction between beneficiaries and non-beneficiaries of public–private partnership initiatives, in particular the Delta Rice Partnership Promotion Project (3PRD) and the Senegal Agricultural Markets Promotion Project (PDMAS). For each HH, we recorded the crops and respective areas cultivated during the three seasons preceding the survey and the income they generated. This helped identify the major crops according to season and net income earned by HHs. Net income were calculated using the following formula:

$$\text{Net income} = (\text{production obtained in kg} \times \text{unit selling price}) - \text{expenses (1)}.$$

The net agricultural income per household was calculated by subtracting total production costs from the gross harvest value. Production costs included all out-of-pocket expenses: seeds, fertilizers, herbicides, mechanical services (tillage and harvesting), and irrigation water fees. Crucially, while hired labor was recorded as a cash expense, family labor was not

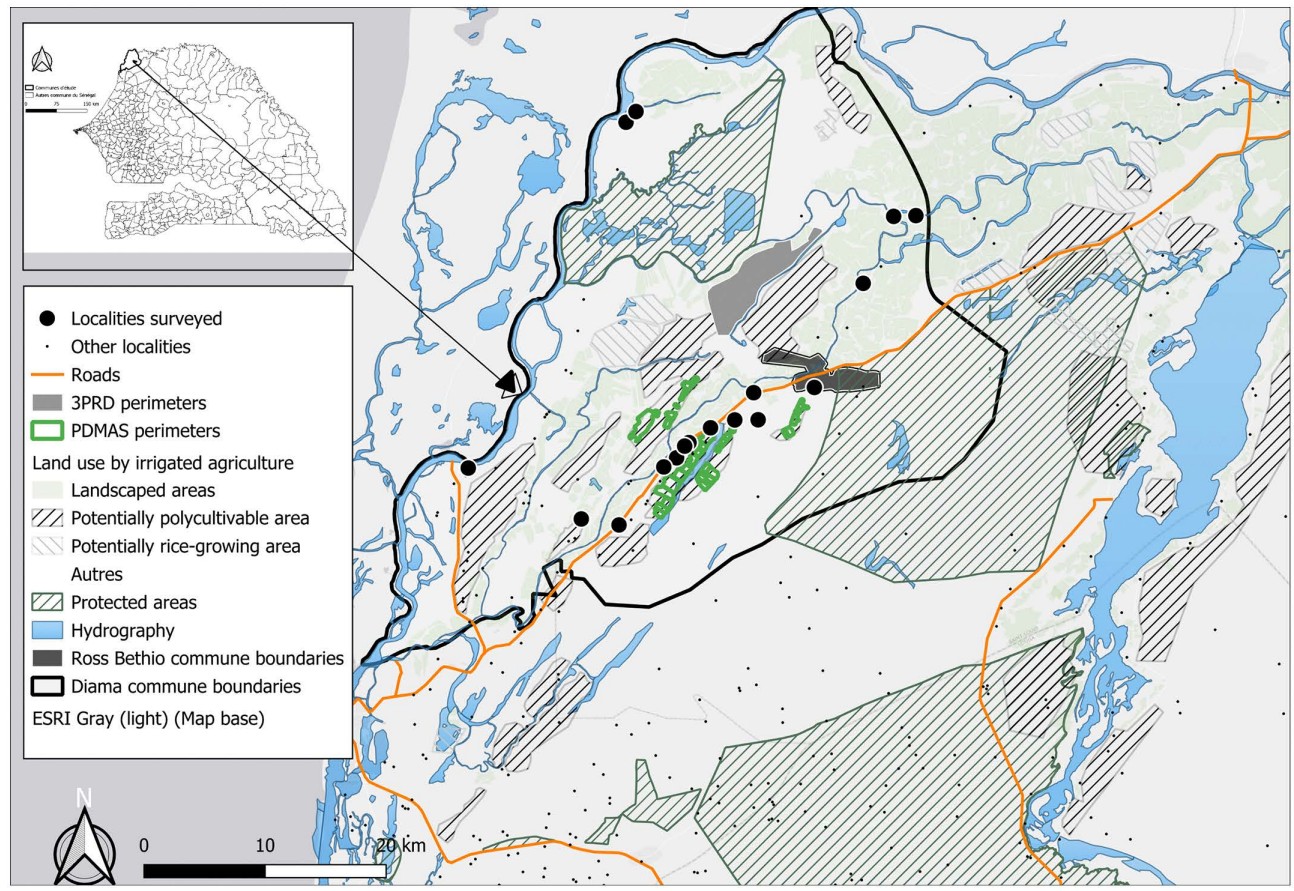

**Fig 1. Location of the villages surveyed.** The map highlights the locations of surveyed villages within the commune boundaries of Ross Bethio and Diama, integrated with existing irrigation perimeters.

assigned a shadow price and was excluded from the cost calculation. Similarly, fixed costs such as equipment depreciation were not included, focusing the analysis on the annual operating margin available to the household.

Fig 1 was produced by the authors using QGIS software. The basemap was derived from OpenStreetMap, and all spatial data were processed and compiled by the authors. The map is original and does not infringe any copyright.

More details on the methodology of this study can be found in [18].

## Measurement of inequalities

Following Bellù and Liberati (2006), we used the Gini coefficient (G) to characterize profit and income distribution over HHs in our sample [19]. The procedure follows 6 steps [20]. In step 1, we sorted the distribution of agricultural profits obtained through our surveys. In step 2, we calculated the distribution of cumulative profits. In step 3, we obtained the cumulative proportion of profits ($q_i$) by dividing each cumulative profit by the total profits. In step 4, we calculated the cumulative proportion of the population ($p_i$) by ranking the individuals in ascending order and assigning rank 1 to the person with the lowest profit and rank "n" to the person with the highest profit, then we divided by n. Step 5 allowed to calculate the area of the polygons Z1, Z2, Z3....Zn. The first is a triangle, the others are trapezoids ($Z_i = \frac{(q_i + q_{i-1})}{2} \frac{(\bar{p}_i - \bar{p}_{i-1})}{}$). Finally, in step 6 we added all areas to obtain the area (Z) under the Lorenz curve ($Z = \sum_{i=1}^{n} Z_i = \frac{1}{2} \sum_i [(q_i + q_{i-1})(p_i - p_{i-1})]$), then we calculated the Gini index (G = 1-2Z). We used the data from our survey to produce the Gini index and Lorenz curve.

 

## Ethics statement

This study did not require approval from the National Ethics Committee for Health Research of Senegal (CNERS), as it focused exclusively on agricultural practices and livelihoods and did not involve the collection of sensitive personal or health-related data. Participation in the survey was entirely voluntary. Prior to each interview, participants were informed of the objectives of the study, the type of information to be collected, and their right to withdraw at any time without any consequence. Verbal informed consent was obtained from all participants and was documented by the enumerators on the survey forms before proceeding with the interviews. This consent procedure was approved by the implementing research institution.

## Result and discussion

### A strong inequality of land owned by households

The 326 households surveyed work on average 7.4 ha (Table 1) with a standard deviation of 24.3. Around 72% of HHs work less than 5 ha, 25% less than 1 ha; and 38% 1–2 ha and 37% 2–5 ha. HHs that own between 5 ha and 20 ha represent 21% of respondents. In this category, 64.5% work areas that are between 5 and10 ha and 35.5% 10–20 ha. Only 7% of the HH surveyed work on 20 ha or more. In this class, 84% of respondents work on between 20–100 ha, whereas some 16% hold 100 ha or more. Unlike the majority of farmers, these large-scale operators are primarily wealthy individual investors, including high-ranking government officials, religious dignitaries, and urban professionals (such as businessmen and physicians). Their presence highlights a significant trend of "absentee" or "investor" farming, where urban capital is used to secure large landholdings, further exacerbating the Gini coefficient for land distribution.

### A strong predominance of rice and hot off-season cropping

Usually, three growing seasons (table 2) are considered in the Senegal River Delta and Valley: the hot off-season (HOS), the rainy season (Winter) and the cold off-season (COS). The rainy season rarely exceeds 3 months, it starts between the end of June and the beginning of August and ends between mid-September and mid-October with a mean temperature range of 23°C min and 35°C max, and high relative humidity. The cold off-season goes from mid-November to the end of February (15–25°C mean daily temperature) with sometimes low humidity due to dry and strong winds. Finally, the hot off-season takes place between March and June, with mean daily temperatures varying between 25° and 40°, that can reach a maximum of 45°C [21].

Survey results show strong concentration of cropping activities on the hot off-season campaign whereas few producers cultivate during the other seasons. 84% of producers cultivate during the hot off-season, compared to only 21% and 26% respectively for the rainy season and cold off-season campaigns (Fig 2).

Table 1. Agricultural areas owned by households (N = 326).

| Farm size | | |
| --- | --- | --- |
| Less than 5 ha | Between 5 ha and 20 ha | 20 ha and more |
| 72% | 21% | 7% |

Table 2. Agricultural seasonal calendar.

| Season | Months | Main Crops | Climate Characteristics |
| --- | --- | --- | --- |
| Hot off-Season | March – June | Rice, onion | High temperatures, high evapotranspiration |
| Rainy Season | July – October | Rice, onion | High humidity, rainfall (300–400 mm) |
| Cold off-Season | November – February | Vegetables, Wheat | Cooler temperatures (15°C – 25°C) |

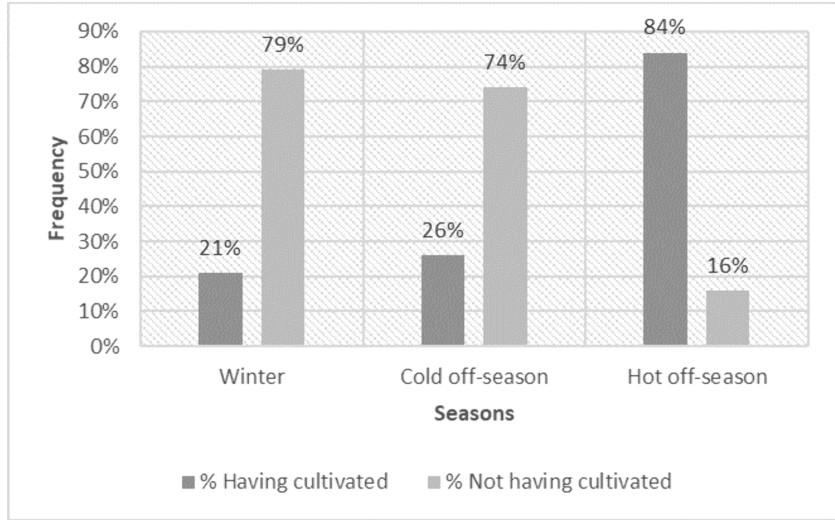

**Fig 2.  Percentage of farmers who cultivate according to season (N = 326).** The chart illustrates the frequency of cultivation versus land dormancy across the rainy season (wintering), cold off-season, and hot off-season (N = 326).

In terms of area held by the producers surveyed, 67.75% is planted during the hot off-season, 12.54% in the rainy season and only 7.24% during the cold off-season (Table 3).

Depending on the season, a variety of crops are grown, including rice, tomatoes, onions, okra, eggplant, potatoes, cassava, watermelon, peanuts, pepper, cabbage, and cucumbers. However, despite the potential of hydro-agricultural developments that would, in theory, allow for three harvests per year, rice remains the dominant crop as illustrated in Fig 3 below. This dominance is intrinsically linked to the relative importance of the hot off-season (HOS), which accounts for 67.75% of total cultivated area (Table 2).

During the hot off-season, rice occupies 86% of all cultivated plots compared to 7% for onion. Other crops occupy smaller shares, rarely exceeding 1%. Rice also covers the highest surface in winter with 61% of plots cultivated, followed again by onion (10%), bell pepper (7.30%), eggplant (6%), watermelon (6%) as well as peanut, cabbage and tomato which each representing 2.40%. On the other hand, with a relative cover of 46.51% of plots, onion constitutes the first crop cultivated during the cold off-season, followed by rice and tomato with 26.35% and 7.75%, respectively. During the same season, some farmers plant bell pepper (4.65%), peanut (3.10%), eggplant (3.87%) as well as cucumber (3%). Few producers cultivate okra (2.32%), watermelon (0.77%), or cassava (1.55%). Thus, rice remains the most cultivated crop in the studied areas with a much higher frequency in the hot off-season than during the other seasons.

## Unequal distribution of agricultural profits

Table 4 shows that the surveyed households achieved an overall profit of 1,098,071,936 FCFA over all three campaigns. Some 77.01% of this profit was produced during the hot off-season campaign, compared to 16.12% for the winter and 6.86% for the cold off-season, resp.

**Table 3.  Share of cultivated areas per season in the total area owned by respondents (N = 326).**

| Variables | Number of producers | Total number of plots | Total Area (ha) | Area cultivated in hot off-season | Cultivated area in winter | Cultivated area in cold off-season |
|---|---|---|---|---|---|---|
| Effective | 326 | 549 | 2408 | 1631.63 | 302.16 | 174.46 |
| Share of total area (%) | | | | 67.75 | 12.54 | 7.24 |

**Fig 3. Frequency of crops in plots according to the growing season.** Distribution of rice and various horticultural crops across the three annual growing cycles (N = 326).

**Table 4. Net households income during the different campaigns (FCFA) (N = 326).**

| Statistics | Campaigns | | | |
|---|---|---|---|---|
| | Hot off-season | Rainy season | Cold off-season | All three campaigns |
| Average (FCFA) | 2,593,993 | 543,097 | 231.228 | 3,368,319 |
| Median (FCFA) | 358,300 | 0 | 0 | 578,000 |
| Minimum (FCFA) | −1,800,000 | −3,250,000 | −2,456,000 | −4,776,945 |
| Maximum (FCFA) | 145,000,000 | 85,400,000 | 10,320,000 | 230,400,000 |
| Total net income in FCFA In Euro | **845,641,731** | **17,7049,755** | **75,380,450** | **1,098,071,936** |
| % of net income | **77.01%** | **16.12%** | **6.86%** | **100%** |

On average, HHs that cultivated during the hot off-season obtained an overall profit of 2,593,993 FCFA (€ 3,956,) with a median of 358,300 FCFA (€ 547). In the winter and cold off-season, this average profit is respectively 543,097 FCFA (€ 828) and 231,228 FCFA (€ 353) with zero medians. The substantial losses reported by some households partly account for these differences. Indeed, losses were relatively lower in the hot off-season-1,800,000 FCFA (− € 2,745) compared to the cold dry season (−2,456,000 FCFA) (− € 2,745) and highest during the rainy season, which recorded a maximum loss of-3,250,000 FCFA (− € 4,956).

Even though overall there are more producers who are doing well (since average profits achieved during the three campaigns is 3,368,319 FCFA (€ 5,136) per HH with a median of 578,000 FCFA (€ 881), there is a high concentration of agricultural income in a minority of HHs while others generate only low profits. To verify this result, we used the Lorenz income distribution curve (Fig 4, A, B, C, D) as well as the Gini index (Table 5).

Fig 4 above illustrate the large inequality that exists between agricultural profits obtained by HHs regardless of the campaign. Indeed, in Fig 4-D which regroups all seasons, the cumulative share of profits obtained during the 3 campaigns is negative for 43% of HHs whereby 93% of the latter group obtain less than 20% of these profits. There is only a small number of less than 7% of producers who hold the bulk of the cumulative share of profits made by HHs for all three grow-ing seasons together. This disparity is confirmed by a high Gini coefficient of 0.88 (Table 5).

Table 5 above shows that the Gini coefficient is very high when the percentage of HHs having produced is low. Indeed, it is 0.94 for the rainy season; 0.91 for the cold off-season and 0.85 for the hot off-season, while respectively 21%, 26%

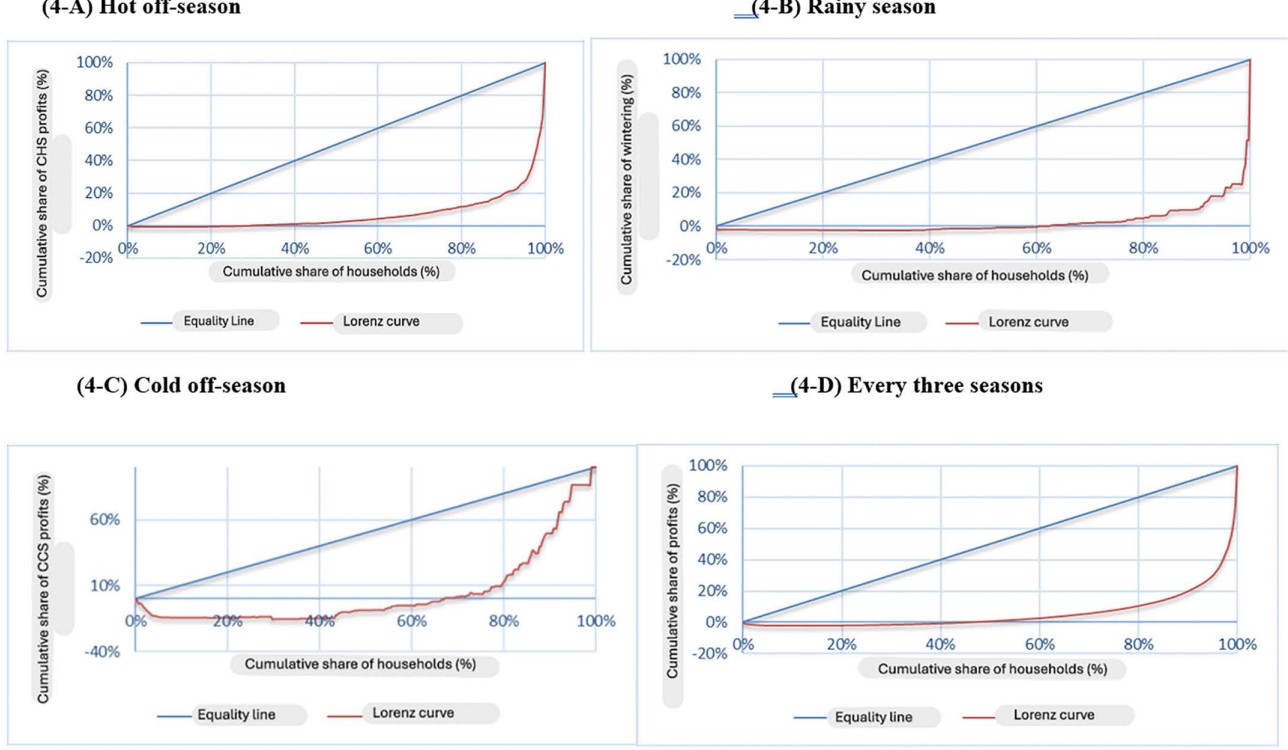

**Fig 4. Lorenz curve of the distribution of agricultural profit realized by household during the campaigns.** (4-A) Hot off-season, (4-B) Rainy season, (4-C) Cold off-season, and (4-D) Aggregated annual profit for all three seasons (N = 326).

**Table 5. Decomposition of inequalities in agricultural profits according to seasons (N = 326).**

| Seasons | Hot off-season (HOS) | Rainy season | Cold off-season (COF) | Gini coefficient of agricultural profits for all three seasons |
|---|---|---|---|---|
| Gini coefficient of agricultural profits for the season | 0.85 | 0.94 | 0.91 | 0.88 |
| Number of households having cultivated | 274 | 70 | 85 | |
| Percentage of households having cultivated | 84% | 21% | 26% | |

and 84% of HHs were actively growing crops during these seasons. Whatever the season, the Gini coefficient exceeds 0.8, which shows a strong inequality between the respondents in terms of agricultural profits.

## Discussion

Several factors can explain the inequalities in farm incomes as reported by the Gini coefficients. Among these factors, the area actually sown is the most decisive. Previous analysis by Sylla et al. (2021) already highlighted significant inequalities in land access among Delta farmers, showing that those cultivating larger areas consistently achieve higher incomes. The same authors found that 66% of Delta irrigators cultivate at most 2 ha during the three seasons and earn less than 1 million FCFA per year from agriculture. On the other hand, 1.8% of HHs cultivate more than 90 ha per year and generate more than 50 million FCFA (€ 22867.4) in profits and have a higher standard of living than the others. Finally, 32.2% of

HHs cultivate 8 ha per year and earn 3.5 million FCFA (€ 5335.72). Their standard of living is average compared to the poorest and richest groups [13]. The small areas cultivated by small-scale irrigators in Sub-Saharan Africa limit their ability to escape poverty. Unlike what happens with large producers, their farms do not allow them to acquire the necessary machinery to make the best of the irrigation offered, a situation that accentuates inequalities [6].

Elouaamari and Lagarde (2025) emphasize that irrigation development in the Senegal River Delta has reproduced and, in some cases, deepened existing social and territorial inequalities. The authors show that access to irrigated land and water is highly uneven, shaped by historical policies, socio-economic status of farmers, and institutional arrangements. Large-scale irrigation schemes, often managed by public or private entities, tend to favour wealthier or better-connected farmers who can afford inputs, equipment, and administrative fees. In contrast, smallholders and young farmers are often relegated to marginal lands or excluded from formal irrigation networks [22].

Furthermore, in Senegal, agricultural income inequality does not only concern irrigated areas. It is a phenomenon that is also widespread in rainfed agriculture areas. The study by Faye et al. (2019) in the so-called Peanut Basin of Senegal also shows a strong income inequality between farmers. As in the delta, the rich farmers of the Peanut Basin cultivate three times more land than the poorest. Inequalities in area cultivated are also seen in the agricultural income generated [23].

The extreme Gini coefficients we report reflect the "high-investment, high-risk" nature of irrigation in the Delta, a phenomenon also described in the Office du Niger [24]. Our findings confirm the emergence of a dualistic agricultural model where a small elite of urban-based investors composed of high-ranking officials, religious dignitaries, and business professionals displaces traditional family farming. This phenomenon is not unique to Senegal but represents a regional trend in high-value irrigation frontiers. As noted by Jayne et al. (2016), the rise of medium and large-scale "urban farmers" is fundamentally reshaping land distribution patterns in Africa, often at the expense of local equity [25]. Research by Bélières et al. (2014) demonstrates that when entry costs, such as irrigation fees and mechanization, exceed the liquid capital of family farms, irrigation acts as a powerful socio-economic differentiator rather than a poverty reduction tool [26].

Other studies conducted in developing countries show that even though irrigation is an effective lever for dealing with climate change [9] and reduce poverty [27,28], it often accentuates pre-existing inequalities. These inequalities are usually due to uneven access to irrigated land [11] and the impact of low agricultural production on the income of households that depend mainly on agriculture. The study by Adeyemi et al. (2018) on poverty and income inequality in southwest Nigeria evidences large disparities HHs, especially those headed by women who tend to be poorer [29]. The work of Fischer et al. (2022) reports that the development of irrigation is often accompanied by an increase in inequalities which mainly affects the poorest farmers who are often weakened by the lack of access to cultivable land and poor agricultural campaigns [30].

Similar to our findings in the Senegal River Delta, irrigators in semi-arid India with larger farms generate higher incomes, while small-scale irrigators struggle with modest profits, further deepening inequalities [11]. The same situation has been described in Brazil where farms larger than 1,000 ha occupy more than 40% of agricultural space with wealthy owners [31].

The persistence of inequalities among irrigators can be explained by the fact that even though agricultural growth reduces rural poverty faster than growth in other sectors, it is less effective in reducing inequalities [32]. The studies of Marrero and Servén (2021) and Min and Rao (2023) also confirm that growth, even if it causes a reduction in poverty, often leads to an increase in inequalities [33,34].

The second factor, explaining inequalities among the households surveyed, is the priority given to rice cultivation over other crops, and the hot off-season over other campaigns. However, the hydro-agricultural developments carried out in the Senegal River Delta allow HHs to cultivate at least twice a year. They offer the possibility to grow rice during both the hot off-season and rainy season, with additionally vegetables, such as onions, tomatoes, okra, eggplant, peppers, cabbage and cucumber, during the cold off-season. However, while rice cultivation during the hot off-season benefits from oversight by state agencies and financial support from the Agricultural Bank and credit unions, households cultivating during the

rainy and cold off-seasons must secure their own financing for market gardening activities. Only a small number of tomato producers succeed in obtaining production contracts with agro-industrial firms [35].

Rice cultivation in the hot off-season is usually carried out collective hydro-agricultural plots, set up by SAED with adequate infrastructure. SAED is the public institution historically responsible for the development and organization of irrigation in the Senegal River valley and delta. SAED established a great number of collective perimeters equipped with adequate irrigation and drainage infrastructure, where most of the hot off-season rice cultivation (HOS) takes place. Each participating HH has at least one plot and is well organized. On the other hand, rainy and cold off-season crops are usually grown in Private Irrigated schemes (PIP), owned by a minority of farmers, characterized by basic facilities, their distance from the farmers' homes and a lack of drainage which often leads to a progressive salinization of the land. Finally, the majority of HHs starts the hot off-season campaign late because of their dependence on credit to be able to buy seed and inputs. Others are late before harvesting due to the unavailability of combine harvesters. These two factors lead to an overlap between the end of the hot off-season and the beginning of the winter season. This situation does not allow the majority of farmers to finish the hot off-season nor start the rainy season campaigns on time. Moreover, the start of the rainy season is characterized by muddy soil conditions that slow down transport of harvested products, and land preparation, respectively. This institutional and technical asymmetry of the organization of irrigation in the delta is also described by Elouaamari and Lagarde (2025): the design of hydraulic infrastructures and water allocation systems prioritizes rice cultivation and production goals aligned with national food policies, rather than local livelihood diversification. This "top-down" production model has created dependency on centralized management structures, limiting farmers' autonomy and adaptive capacity. Moreover, inequalities are reinforced by differences in access to crop advice, credit, and market opportunities, which contribute to the persistence of a dual agricultural system, one modern and capital-intensive, and the other traditional and precarious [22].

Vegetables are most often planted during the hot off-season, which involves many risks. HHs often struggle to find credit, are not protected from straying livestock and often have difficulties selling onions. Competition from other horticultural production areas, especially the coastal Niayes area that stretches from Dakar to Saint Louis and which accounts for nearly 80% of Senegal's horticultural production [36], further exacerbates the already saturated market and price volatility. Studies in the Niayes reveal similar structural constraints despite higher productivity levels: farmers there also grapple with increasing land pressure, water salinization, and the growing cost of inputs, which threaten the sustainability of this key horticultural hub [1]. Moreover, the lack of adequate storage, cold chain infrastructure, and marketing cooperatives leads to significant post-harvest losses across all production zones, with up to 30–40% of harvests getting lost annually [5].

Added to this, are the problems of rotting produce due to the lack of storage facilities. Surprisingly, the negative incomes in our surveys are higher among those who practice market gardening. The higher the losses, the more difficult it is for households to pay off their debts and start a new season. Similar results have been reported from other countries such as Ghana, where irrigators struggle to earn much income because of low cultivated surface areas and post-harvest losses [29]. In Nigeria, findings from Kolawole et al. (2020) indicate that irrigation can increase poverty if HHs experience poor agricultural seasons, problems marketing their produce, and post-harvest losses that add to the high operating and maintenance costs of irrigation equipment [8]. We can also add to this the soil salinization and drop in yields or abandonment of the land that this can cause. In our sample, we found that even during the hot off-season, which is the peak growing season, 32% of the land is not planted, mainly due to these problems. For smallholders, the financial inability to invest in soil restoration or high-cost irrigation during the dry season exacerbates this phenomenon, leaving a third of the delta's productive potential dormant.

The problems reported in the Delta are also found throughout the Senegal River Valley. According to recent SAED data (2024), the analysis of agricultural trends in the Senegal River Valley highlights a significant increase in rice cultivation areas, from 23,197 ha in 2002/03–75,499 ha in 2019/20 (Fig. 5A). However, this expansion is accompanied by a

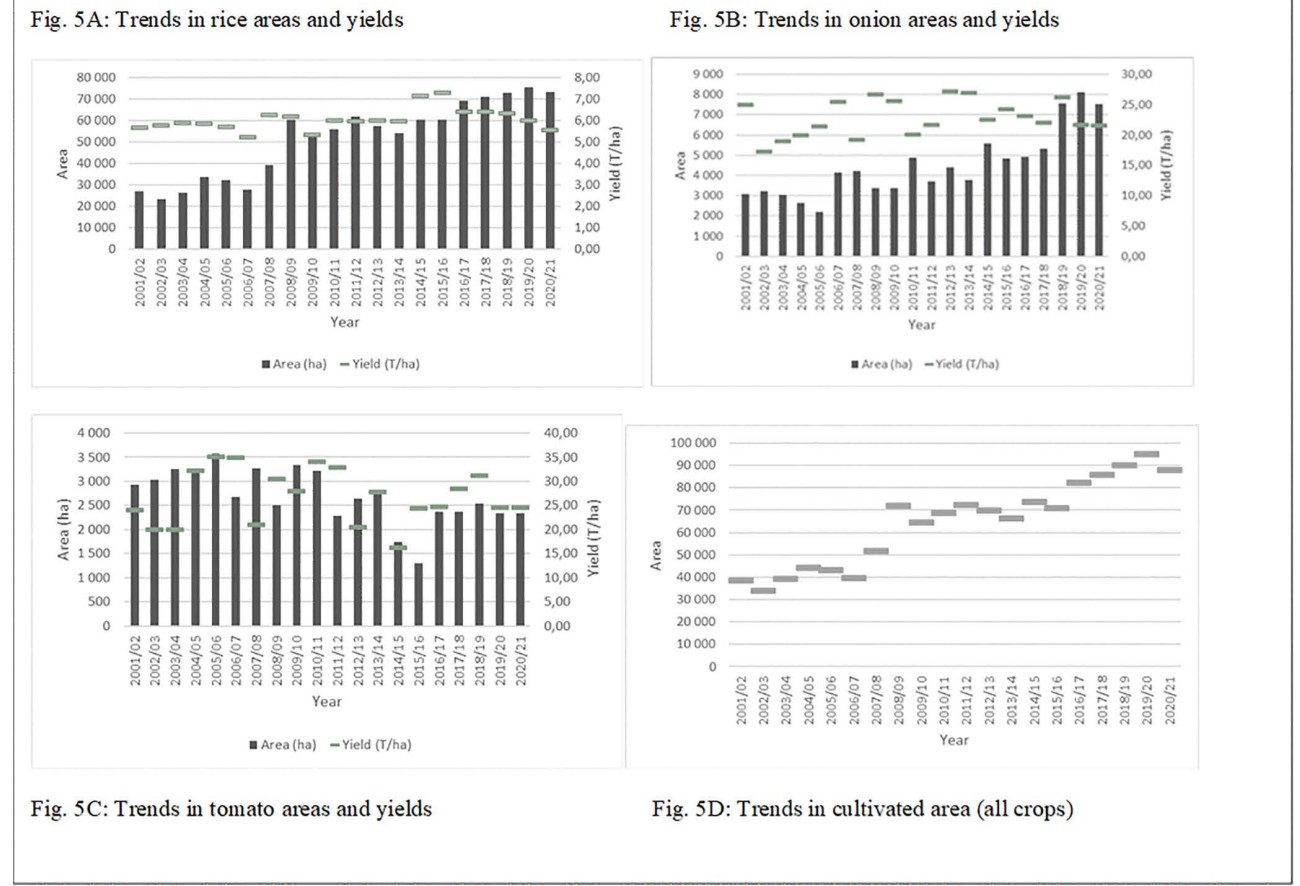

**Fig 5. Trend in crop areas and yields in the Senegal River Valley from 2001 to 2021.** (5-A) Rice, (5-B) Onion, (5-C) Tomato, and (5-D) Total annual cultivated area for all crops.

continuous decline in yields, which dropped from 7.28 T/ha in 2015/16 to 5.55 T/ha in 2020/21. In contrast, tomato cultivation has been in decline for over a decade, with decreasing cultivated areas and unstable yields ranging between 15 and 35 T/ha (Fig 5C). On the other hand, onion farming has expanded significantly, with cultivated areas increasing from 3,093 ha in 2000/01–8,120 ha in 2019/20 (Fig 5B), although yields remain highly variable (17–28 T/ha).

The significant decline in rice yields observed highlights the fragility of the Delta's agricultural model. This 24% drop is not merely a statistical fluctuation but the result of deepening structural constraints. Chief among these is soil salinization caused by inadequate drainage, which has led to the gradual abandonment of land or reduced productivity in older schemes. Furthermore, the rising cost of inputs following the liberalization of the sector has forced many low-income households to adopt sub-optimal fertilization strategies. This is compounded by the obsolescence of hydraulic infrastructure, where silted canals and failing pumps prevent precise water management. These factors create a vicious cycle: falling yields reduce the surplus available for reinvestment, further widening the inequality gap between smallholders and well-capitalized investors who can maintain private pumping systems and high input levels

Ultimately, rice and onions are the primary drivers of cultivated land expansion, while other crops such as tomatoes and maize show a declining trend. Despite this expansion, the total cultivated area remains well below its potential. In 2019/20, only 90,129 hectares were farmed (Fig. 5D), whereas SAED estimates that approximately 240,000 hectares are suitable

for irrigation [37].These findings confirm the underutilization of agricultural land and highlight the need to further develop rice cultivation while reducing post-harvest losses in vegetable production to stabilize incomes.

This finding is widespread across Senegal and partly explains the government's decision to develop the *National Strategy for Food Sovereignty (SNSA)* for the 2024–2028 period, which was adopted in January 2023. Among the corrective measures implemented by the State, two stand out as transformative in reshaping the country's agricultural landscape and improving people's access to staple foods. The first involves the introduction *rainfed rice cultivation* into production systems and areas that had previously been excluded from rice farming. The second consists of *introducing wheat production* in Senegal, with the dual objective of reducing the country's trade deficit and wheat imports, and ensuring better access and availability of wheat in local markets. The government's ambitions under this new policy framework are particularly significant, aiming by 2025 to achieve full domestic coverage of rice needs and 42% coverage in locally produced wheat, alongside self-sufficiency in other key crops such as onions, potatoes, and carrots [38]. Such efforts would increase farmers' incomes, reduce inequalities among producers, and contribute to Senegal's goal of achieving food sovereignty.

## Conclusion

Irrigation in the Senegal River Delta is characterized by large inequalities in land use (and ownership) and incomes. A significant share of agricultural income is obtained by a minority of producers who own large areas of land. Small producers, who are the majority, cultivate small areas that do not allow them to generate significant profits and thus cannot accumulate any significant amount of money. Despite the availability of enough water for continuous irrigation and favourable climatic conditions allowing for three seasons cropping, more than half of HHs only grow crops during the hot off-season. The strong interest in the hot off-season can be attributed to its more structured organization and the comparatively lower level of risk it entails relative to the other seasons. The absence of agricultural activity during the other seasons is largely driven by organizational and technical constraints, leading to substantial losses in potential income. Beyond exacerbating poverty and inequality, this underutilization of arable land reduces domestic food production and increases reliance on imports, thereby further deepening the country's structural trade balance deficit. This study examined inequality at the household level; however, other dimensions of inequality might emerge if variables such as gender, age, or ethnicity of producers were considered. This represents a limitation of the current analysis and an opportunity for future research to explore these additional factors.

To address the high rate of land dormancy and the extreme inequalities identified in this study, One of the implications of the analysis is implementation of results-based performance contracts. This policy shift would move land governance from a purely administrative model to one based on productivity and accountability, as suggested by Garces-Restrepo et al. (2007) [39]. Beyond the technical optimization of irrigation, the future of the Senegal River Delta depends on a balance between large-scale investment and social equity. Within the framework of the 'Senegal 2050' benchmark, the country's new economic development agenda, the State is banking on the creation of large-scale agropoles to boost agricultural productivity and ensure national food sovereignty. However, to be truly transformative, this new strategic agenda must better integrate smallholder farmers and impose strict results-based mandates on large-scale operators, defining a minimum cropping intensity to be achieved by all actors. Failing to enforce such performance standards risks perpetuating the structural problems described in this study: widening inequalities, land underutilization, and the continuous postponement of food sovereignty goals.

## Supporting information

**S1 Data. Anonymized household survey data.** This file contains the raw data used for land and income inequality calculations, including Gini coefficients and Lorenz curves.
(XLSX)

**S2 Data. Dataset of cultivated areas and agricultural production in the Senegal River Valley (2001–2021).** This Excel file contains the raw data used to analyze land distribution, cropping intensities, and yield trends presented in this study.
(XLSX)

## Acknowledgments

We would like to thank the farmers of the Senegal River Delta who answered our questions, and the authorities of the municipalities of Diama and Ross-Béthio who provided us with data on the population and owners of agricultural plots. We would also like to thank the authorities of the SAED for providing us with data on agricultural production and hydro-agricultural developments of the Delta and the entire Senegal River Valley.

## Author contributions

**Conceptualization:** El Hadji Malick Sylla, Bruno Barbier, Boubacar Ba.

**Data curation:** El Hadji Malick Sylla.

**Formal analysis:** El Hadji Malick Sylla.

**Investigation:** El Hadji Malick Sylla.

**Methodology:** El Hadji Malick Sylla, Bruno Barbier.

**Supervision:** El Hadji Malick Sylla, Bruno Barbier, Patrick Van Damme, Sidy Mohamed Seck.

**Validation:** El Hadji Malick Sylla.

**Writing – original draft:** El Hadji Malick Sylla.

**Writing – review & editing:** El Hadji Malick Sylla, Bruno Barbier, Boubacar Ba, Khalifa Diop, Patrick Van Damme, Sidy Mohamed Seck.

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
