## [Decision Letter · Decision Letter 0]

2 Oct 2025

Dear Dr. Sylla,

Thank you for submitting your manuscript to PLOS ONE. After careful consideration, we feel that it has merit but does not fully meet PLOS ONE’s publication criteria as it currently stands. Therefore, we invite you to submit a revised version of the manuscript that addresses the points raised during the review process.

We look forward to receiving your revised manuscript.

Kind regards,

Annesha Sil, Ph.D.

Staff Editor

PLOS ONE

Journal Requirements:

1. Please ensure that your manuscript meets PLOS ONE’s style requirements, including those for file naming. The PLOS ONE style templates can be found at

https://journals.plos.org/plosone/s/file?id=ba62/PLOSOne_formatting_sample_title_authors_affiliations.pdf....

2. Please provide details regarding participant consent. In the ethics statement in the Methods and online submission information, please ensure that you have specified (1) whether consent was informed and (2) what type you obtained (for instance, written or verbal, and if verbal, how it was documented and witnessed). If your study included minors, state whether you obtained consent from parents or guardians. If the need for consent was waived by the ethics committee, please include this information.

a. You may seek permission from the original copyright holder of Figure 1to publish the content specifically under the CC BY 4.0 license.

5. Please remove your figures from within your manuscript file, leaving only the individual TIFF/EPS image files, uploaded separately. These will be automatically included in the reviewers’ PDF

Reviewers' comments:

Reviewer’s Responses to Questions

**Comments to the Author**

1. Is the manuscript technically sound, and do the data support the conclusions?

Reviewer #1: Partly

2. Has the statistical analysis been performed appropriately and rigorously?

Reviewer #1: Yes

3. Have the authors made all data underlying the findings in their manuscript fully available?

Reviewer #1: No

4. Is the manuscript presented in an intelligible fashion and written in standard English?

Reviewer #1: No

Reviewer #1: This manuscript addresses a relevant and timely topic: inequalities in land access and agricultural income among irrigated farmers in the Senegal River Delta. The data collected are valuable, the use of Gini and Lorenz analyses is appropriate, and the focus on an under-researched context adds originality to the work.

To fully meet the standards of a scientific publication in PLOS ONE, the manuscript would benefit from major revisions. While the study is well documented, its structure and tone currently resemble a technical report more than a scholarly article. Enhancing the clarity of the writing, deepening the analytical engagement, and strengthening the contextual framing — particularly with regard to institutional and policy dimensions — would significantly improve the paper. The following comments are offered with the intention of constructively supporting the authors in enhancing the overall quality and impact of their work.

*The manuscript requires careful proofreading by a native English speaker or language editor. Issues such as incomplete sentences (e.g.,Discussion p. 9), repeated words (e.g., “UsuallyUsually”, p. 4), grammatical error (e.g., disapointed/disappointed) and unclear transitions affect readability and precision.

*The text briefly mentions Senegal’s goal of food sovereignty by 2029 but does not elaborate on current national policies or programs related to rice production. A more developed discussion of policy efforts and institutional strategies would help situate the findings within present-day challenges and debates.

*SAED is mentioned but not defined or contextualized. Clarifying its status (e.g., government agency) and role in irrigation policy would be useful for non-specialist readers. Additionally, the ISRA — a major actor in agricultural research in Senegal — is not mentioned at all. If relevant, its role should be acknowledged to provide a more complete picture of the institutional landscape.

*The paper presents a useful snapshot of inequalities but would benefit from a more dynamic and process-oriented interpretation. For example, exploring how inequalities have emerged or been reproduced over time (e.g., through inheritance, reforms, market access) could deepen the analysis, even with cross-sectional data.

*The analysis remains largely descriptive. To strengthen the paper’s academic contribution, the authors could more explicitly connect their findings to broader theoretical or structural debates on inequality, rural development, and irrigation governance.

*While the quantitative data are valuable, the narrative does not always reference or explain tables and figures clearly (e.g., tables 2 and 3). Improving the integration of visual elements into the text would enhance clarity.

*The study treats households as a homogeneous unit, without exploring variables such as gender, age, or ethnicity, which are often relevant to understanding inequality. While this may be beyond the current scope, acknowledging it as a limitation would strengthen the manuscript.

*In line with PLOS ONE’s open science standards, it would be helpful to indicate whether the dataset can be shared publicly (even in anonymized form), or to explain any constraints to doing so.

I commend the authors for their fieldwork in a challenging context and for addressing a subject of critical importance. With substantial revisions to enhance the manuscript’s clarity, analytical depth, and contextual grounding, this study has the potential to make a valuable contribution to the literature on irrigation and inequality in West Africa. I hope these comments are helpful and supportive of the authors’ continued work on this topic.

.

Reviewer #1: **Yes:** Fagandini Ruiz F.Fagandini Ruiz F.Fagandini Ruiz F.Fagandini Ruiz F.

---

## [Author Response · Author response to Decision Letter 1]

16 Jan 2026

This updated version incorporates all the comments and suggestions provided by the reviewer, whose constructive and insightful feedback has greatly contributed to strengthening the clarity, rigor, and overall quality of the article. We are grateful to the reviewer for their careful reading and thoughtful recommendations. We also sincerely thank the journal for its patience throughout this revision process.

---

## [Decision Letter · Decision Letter 1]

10 Feb 2026

Irrigation and inequalities in the Senegal River Delta

PLOS One

Dear Dr. Sylla,

Thank you for submitting your manuscript to PLOS ONE. After careful consideration, we feel that it has merit but does not fully meet PLOS ONE’s publication criteria as it currently stands. Therefore, we invite you to submit a revised version of the manuscript that addresses the points raised during the review process.

https://journals.plos.org/plosone/s/submission-guidelines#loc-laboratory-protocols. Additionally, PLOS ONE offers an option for publishing peer-reviewed Lab Protocol articles, which describe protocols hosted on protocols.io. Read more information on sharing protocols at . Additionally, PLOS ONE offers an option for publishing peer-reviewed Lab Protocol articles, which describe protocols hosted on protocols.io. Read more information on sharing protocols at . Additionally, PLOS ONE offers an option for publishing peer-reviewed Lab Protocol articles, which describe protocols hosted on protocols.io. Read more information on sharing protocols at . Additionally, PLOS ONE offers an option for publishing peer-reviewed Lab Protocol articles, which describe protocols hosted on protocols.io. Read more information on sharing protocols at https://plos.org/protocols?utm_medium=editorial-email&utm_source=authorletters&utm_campaign=protocols....

We look forward to receiving your revised manuscript.

Kind regards,

Noé Aguilar-Rivera

Academic Editor

PLOS One

Journal Requirements:

Reviewer’s Responses to Questions

**Comments to the Author**

Reviewer #1: All comments have been addressed

Reviewer #2: All comments have been addressed

Reviewer #3: All comments have been addressed

2. Is the manuscript technically sound, and do the data support the conclusions?

Reviewer #1: Yes

Reviewer #2: Yes

Reviewer #3: Yes

3. Has the statistical analysis been performed appropriately and rigorously?

Reviewer #1: Yes

Reviewer #2: Yes

Reviewer #3: No

4. Have the authors made all data underlying the findings in their manuscript fully available?

Reviewer #1: Yes

Reviewer #2: Yes

Reviewer #3: Yes

5. Is the manuscript presented in an intelligible fashion and written in standard English?

Reviewer #1: Yes

Reviewer #2: Yes

Reviewer #3: Yes

Reviewer #1: Thank you for the careful and constructive revision of the manuscript. The authors have addressed all reviewer comments in a substantive way, leading to a clearer, more analytically grounded, and better contextualized paper. I consider the manuscript ready for publication

Reviewer #2: Thank you for submitting this important research on irrigation inequalities in the Senegal River Delta. The study addresses a critical issue affecting agricultural development in West Africa, and your fieldwork with 326 households provides valuable empirical evidence. The use of Gini coefficients and Lorenz curves is appropriate for analyzing income distribution, and the finding that 72% of households cultivate less than 5 hectares while facing significant income deficits highlights serious equity concerns. The revisions have improved the manuscript considerably, particularly the expanded discussion of SAED’s role and the addition of recent policy context. However, there are still some areas where additional clarification or modification would strengthen the paper’s contribution and make it more accessible to international readers.

Comment 1: In the Methods section (pages 10-11, around line 174), you mention that villages were selected based on geographical position, ethnic group, and demographic weight. Could you explain more clearly how you actually chose the 20 specific villages from all possible villages in Diama and Ross Béthio? Did you use a systematic approach, or were there practical considerations that guided this selection? This would help readers understand if your sample might be biased in any particular direction.

Comment 2: On page 11 where you describe the income calculation formula (equation 1), I'm wondering if you could discuss what expenses were included? For example, did you count labor costs, land rental, water fees, or only material inputs like seeds and fertilizers? Also, how did you handle households that use family labor versus hired labor - were these treated differently in the expense calculations?

Comment 3: Looking at Table 1 (page 11), you show that 7% of households farm 20 hectares or more. In the text you mention that some of these have over 100 hectares. Can you provide more detail about this group - roughly how many households fall into this very large category, and are they primarily private companies, cooperatives, or wealthy individual farmers? Understanding who these large operators are would help contextualize the inequality patterns.

Comment 4: The three growing seasons are described on page 12 (around lines 233-238), but the date ranges seem to overlap. For instance, you say hot off-season is February-June and also March-June in different places. Could you provide a clearer table or figure showing the exact months for each season and how they connect? This would help readers unfamiliar with the region understand the agricultural calendar better.

Comment 5: In Table 2 (page 12), you show that 67.75% of cultivated area is in hot off-season, but then you mention that 32% of land is not exploited even during this peak season (page 13, discussion section). Could you clarify this apparent contradiction? Are you referring to different land categories, or is there a calculation I'm misunderstanding?

Comment 6: Your Gini coefficient results in Table 4 (page 14) are quite high, ranging from 0.85 to 0.94. How do these values compare to other agricultural regions in Senegal or similar irrigation schemes in West Africa? Providing some comparative context would help readers assess whether the Delta situation is exceptionally unequal or reflects broader regional patterns.

Comment 7: In the discussion section around page 15, you cite several studies from India, Nigeria, and Brazil to support your findings. While these are relevant, I noticed fewer references to other irrigation schemes in francophone West Africa. Are there studies from Mali’s Office du Niger or Mauritania’s irrigation areas that might provide more direct comparisons? If such studies don't exist, mentioning this gap could strengthen your contribution.

Comment 8: Figure 5 presents trends from 2001-2021, showing declining rice yields from 7.28 T/ha to 5.55 T/ha. This is a dramatic drop, but the text doesn't really explain why yields are falling so sharply. Could you add some discussion about the possible causes - is it soil degradation, reduced fertilizer use, water management problems, or other factors? This seems like a critical issue that deserves more attention.

Comment 9: On page 16 in the conclusion, you recommend that the government should "impose a results contract on large producers and define a minimum crop intensity to be achieved by all operators." Could you elaborate on what you mean by "results contract" and how this would work practically? What kind of crop intensity would be reasonable to require, and how would it be enforced given the organizational challenges you've described?

Comment 10: Regarding your data availability statement, you now say data are available "on reasonable request" to the corresponding author. Given that your data appear to be fully anonymized (no personally identifiable information), is there a specific reason why you cannot deposit the dataset in a public repository like Figshare or Zenodo? Making the data more openly available would increase the transparency and potential impact of your work.

Reviewer #3: title doesn't match with the included subject Imported missing data 1-weather condition to be gathered in one table for different seasons 2- type of irrigation system 3-Type of soil under study physical or chemical soil properties 4-howmuch water needed for rice cultivation under irrigation 5- Figures are not clear 6- In figure 2 you write cultivated and compare with those uncultivated .

.

Reviewer #1: **Yes:** Francesca FAGANDINIFrancesca FAGANDINIFrancesca FAGANDINIFrancesca FAGANDINI

Reviewer #2: No

Reviewer #3: No

You may also use PLOS’s free figure tool, NAAS, to help you prepare publication quality figures: https://journals.plos.org/plosone/s/figures#loc-tools-for-figure-preparation

---

## [Author Response · Author response to Decision Letter 2]

25 Mar 2026

Dear Editor-in-Chief,

Please find enclosed our revised manuscript titled "Irrigation and inequalities in the Senegal River Delta", which we are resubmitting for further consideration in PLOS ONE.

We would like to thank the reviewers for their insightful and constructive feedback. We have addressed every comment, ensuring that the socio-economic analysis of the Gini coefficients is now better integrated with the technical and institutional realities of the Senegal River Delta.

The key revisions in this version include:

We believe that these revisions have significantly improved the clarity and academic rigor of our work. We confirm that this manuscript has not been published elsewhere and is not under consideration by any other journal.

Thank you for your time and for the opportunity to revise our work. We look forward to your positive decision.

Sincerely,

---

## [Decision Letter · Decision Letter 2]

1 Apr 2026

Irrigation and inequalities in the Senegal River Delta

PONE-D-25-26975R2

Dear Dr. El Hadji Malick Sylla

We’re pleased to inform you that your manuscript has been judged scientifically suitable for publication and will be formally accepted for publication once it meets all outstanding technical requirements.

Kind regards,

Noé Aguilar-Rivera

Academic Editor

PLOS One

Additional Editor Comments (optional):

Reviewers' comments:

Reviewer’s Responses to Questions

**Comments to the Author**

Reviewer #2: All comments have been addressed

Reviewer #3: All comments have been addressed

2. Is the manuscript technically sound, and do the data support the conclusions?

Reviewer #2: Yes

Reviewer #3: Yes

3. Has the statistical analysis been performed appropriately and rigorously?

Reviewer #2: Yes

Reviewer #3: Yes

4. Have the authors made all data underlying the findings in their manuscript fully available?

Reviewer #2: Yes

Reviewer #3: Yes

5. Is the manuscript presented in an intelligible fashion and written in standard English?

Reviewer #2: Yes

Reviewer #3: Yes

Reviewer #2: (No Response)

Reviewer #3: Thanks for their great effort no additional comments all needed changes were done the authors have added all needed details and made figures more clear

.

Reviewer #2: No

Reviewer #3: **Yes:** Maybelle saad gaballahMaybelle saad gaballahMaybelle saad gaballahMaybelle saad gaballah

---

## [Editor Report · Acceptance letter]

PONE-D-25-26975R2

PLOS One

Dear Dr. Sylla,

I'm pleased to inform you that your manuscript has been deemed suitable for publication in PLOS One. Congratulations! Your manuscript is now being handed over to our production team.

Kind regards,

on behalf of

Dr. Noé Aguilar-Rivera

Academic Editor

PLOS One